# A Highly Conserved Iron-Sulfur Cluster Assembly Machinery between Humans and Amoeba *Dictyostelium discoideum*: The Characterization of Frataxin

**DOI:** 10.3390/ijms21186821

**Published:** 2020-09-17

**Authors:** Justo Olmos, María Florencia Pignataro, Ana Belén Benítez dos Santos, Mauro Bringas, Sebastián Klinke, Laura Kamenetzky, Francisco Velazquez, Javier Santos

**Affiliations:** 1Instituto de Biociencias, Biotecnología y Biología Traslacional (iB3), Departamento de Fisiología y Biología Molecular y Celular, Facultad de Ciencias Exactas y Naturales, Universidad de Buenos Aires, Intendente Güiraldes 2160, Ciudad Universitaria, Buenos Aires C1428EGA, Argentina; justo.olmos388@gmail.com (J.O.); mariaflorenciapignataro@gmail.com (M.F.P.); benitezbelen843@gmail.com (A.B.B.d.S.); lauka@fbmc.fcen.uba.ar (L.K.); 2Departamento de Química Inorgánica, Analítica y Química Física, Facultad de Ciencias Exactas y Naturales, Universidad de Buenos Aires, Instituto de Química Física de los Materiales, Medio Ambiente y Energía (INQUIMAE CONICET), Buenos Aires C1428EGA, Argentina; mbringas@qi.fcen.uba.ar; 3Fundación Instituto Leloir, IIBBA-CONICET, and Plataforma Argentina de Biología Estructural y Metabolómica PLABEM, Av. Patricias Argentinas 435, Buenos Aires C1405BWE, Argentina; sklinke@leloir.org.ar; 4IMPaM, CONICET, Facultad de Medicina, Universidad de Buenos Aires, Buenos Aires C1121ABG, Argentina; 5Instituto de Química Biológica de la Facultad de Ciencias Exactas y Naturales (IQUIBICEN)—(UBA/CONICET), Buenos Aires C1428EGA, Argentina; 6Consejo Nacional de Investigaciones Científicas y Técnicas, Rivadavia 1917, Buenos Aires C1033AAJ, Argentina; 7Departamento de Química Biológica, Facultad de Ciencias Exactas y Naturales, Universidad de Buenos Aires. Intendente Güiraldes 2160, Ciudad Universitaria, Buenos Aires C1428EGA, Argentina

**Keywords:** conformational stability, protein–protein interaction, iron–sulfur cluster assembly, Friedreich’s Ataxia, *Dictyostelium discoideum*

## Abstract

Several biological activities depend on iron–sulfur clusters ([Fe-S]). Even though they are well-known in several organisms their function and metabolic pathway were poorly understood in the majority of the organisms. We propose to use the amoeba *Dictyostelium discoideum*, as a biological model to study the biosynthesis of [Fe-S] at the molecular, cellular and organism levels. First, we have explored the *D. discoideum* genome looking for genes corresponding to the subunits that constitute the molecular machinery for Fe-S cluster assembly and, based on the structure of the mammalian supercomplex and amino acid conservation profiles, we inferred the full functionality of the amoeba machinery. After that, we expressed the recombinant mature form of *D. discoideum* frataxin protein (DdFXN), the kinetic activator of this pathway. We characterized the protein and its conformational stability. DdFXN is monomeric and compact. The analysis of the secondary structure content, calculated using the far-UV CD spectra, was compatible with the data expected for the FXN fold, and near-UV CD spectra were compatible with the data corresponding to a folded protein. In addition, Tryptophan fluorescence indicated that the emission occurs from an apolar environment. However, the conformation of DdFXN is significantly less stable than that of the human FXN, (4.0 vs. 9.0 kcal mol^−1^, respectively). Based on a sequence analysis and structural models of DdFXN, we investigated key residues involved in the interaction of DdFXN with the supercomplex and the effect of point mutations on the energetics of the DdFXN tertiary structure. More than 10 residues involved in Friedreich’s Ataxia are conserved between the human and DdFXN forms, and a good correlation between mutational effect on the energetics of both proteins were found, suggesting the existence of similar sequence/function/stability relationships. Finally, we integrated this information in an evolutionary context which highlights particular variation patterns between amoeba and humans that may reflect a functional importance of specific protein positions. Moreover, the complete pathway obtained forms a piece of evidence in favor of the hypothesis of a shared and highly conserved [Fe-S] assembly machinery between Human and *D. discoideum.*

## 1. Introduction

### 1.1. Iron–Sulfur Cluster Assembly in Mammals

Our laboratory has focused on understanding (at the molecular level) the relationships between structure, dynamics and function of the mitochondrial supercomplex involved in the biosynthesis of iron–sulfur clusters [Fe-S]. This biosynthetic pathway is vital for various enzymatic functions such as respiratory chain complexes [1,2], aconitase and succinate dehydrogenase [3], and in other critical processes such as DNA repair [4] and chemical modification of transfer RNAs [5,6]. Therefore, several biological activities depend on [Fe-S].

The mammalian mitochondrial supercomplex for assembling [Fe-S] is made up of several proteins [7,8,9,10]. Among them, cysteine desulfurase NFS1 is one of the key enzymes and is encoded by nuclear DNA [11]. NFS1 is translated as a 50 kDa precursor protein, imported into the mitochondrial matrix, and processed to give its mature dimeric form (2 × 44.5 kDa). It works as a hub collecting many other proteins on its surface; more specifically, NFS1 is a pyridoxal-phosphate (PLP) dependent enzyme, which catalyzes the desulfurization of *L*-cysteine generating the precursor sulfide attached as a persulfide group to a Cys residue (Cys-S-SH) and L-alanine. This key Cys residue of NFS1 is situated in a mobile loop (CYS loop) that is involved in the subsequent persulfide group transference. One of the proteins that participate in this process is iron–sulfur cluster assembly enzyme (ISCU), an essential protein that is specifically the scaffold that supports the assembly of the [2Fe-2S] cluster in its assembly site, which is made up of three cysteines, one histidine and one aspartic residue. Crystallographic data show that the ISCU active site is in close contact with the Cys loop from the NFS1 active site [7]. For the human molecular machinery, it has been exquisitely demonstrated that Cys69, Cys95, Asp71 and His137 are indeed involved in iron(II) binding [12], whereas the third Cys residue (Cys138) functions as the acceptor of the persulfide group also needed for [2Fe-2S] assembly. The [2Fe-2S] cluster is coordinated by Cys69, Cy95, His137 and Cys138. It has been reported another, lower affinity, iron binding site in ISCU although its role in iron–sulfur cluster biogenesis remains to be analyzed [12,13]. It should be mentioned that once mounted on the ISCU assembly site, the [2Fe-2S] group is transferred to other proteins. In this process, Asp71, contributes to destabilization and the subsequent release of the cluster [14,15].

Another essential protein of the complex in mammals is frataxin (FXN), which is an activator of NFS1. FXN is also nuclear-encoded protein that positively modulates desulfurase and cluster formation activities [16,17,18]. Since FXN binds iron with moderate K_D_ [19], it was believed that this protein might participate by providing the metal ions necessary for the biosynthesis of the groups [20], however, there is no significant experimental support for this model. Current experimental evidence indicates that FXN binds iron in the context of the protein complex and that FXN could release the metal ion by cysteine (the substrate) and the natural reducing agent FDX2 [20]. On the other hand, Gervason and coworkers reported valuable evidence that FXN is not necessary for iron insertion in ISCU and showed that ISCU per se exhibits the iron binding capability [12]. Moreover, recent studies indicate that FXN works as a kinetic activator of the metal-dependent persulfide transfer process from NFS1 to ISCU [12,18,21], and part of FXN surface specifically fits on the ISCU assembly site [10].

In humans, FXN is imported into the mitochondrial matrix as a 210-residue precursor. It is processed in two steps, giving rise first to an intermediate form (residues 41 to 210) and then the mature FXN form 81–210 (14.2 kDa).

It should be noted that the alteration in the expression or the presence of point mutations in the proteins involved affect their functionality, resulting in human diseases. In this regard, the decrease in the expression of FXN or ISCU, or a decreased functionality of these proteins, results in serious diseases: Friedreich’s Ataxia (FRDA) and ISCU myopathy, respectively [22,23]. In addition, it was shown that mutation of NFS1 results in a novel autosomal recessive mitochondrial disease characterized by the respiratory chain complex II and III deficiency, multisystem organ failure and abnormal mitochondria [24].

In particular, in FRDA, ~95% of the individuals are homozygous for a GAA repeat expansion in the first intron of the FXN gene that produces a reduction of the FXN expression. The remaining individuals with FRDA (~5%) are compound heterozygous for a GAA expansion and a mutation in the second allele of FXN yielding a premature stop codon, a frame shift or a point mutation [25]. Structural and functional studies suggest that FXN interacts simultaneously with ISCU and with both NFS1 subunits. The interaction of FXN and ISCU is mediated by the β-sheet platform of FXN.

On the other hand, the NFS1 desulfurase is stabilized in its active and dimeric conformation by a very small protein of 10.5 kDa called ISD11, which is an essential protein in eukaryotes but that is completely absent in Bacteria and Archaea. It was demonstrated that ISD11 and ACP (mitochondrial acyl carrier protein) give rise to a and heterodimer that once formed interacts with NFS1 stabilizing the active form of the enzyme [26,27].

More importantly, the structure of the supercomplex (NFS1/ACP-ISD11/ISCU/FXN)_2_ has been recently obtained by cryo-electron microscopy and this puts FXN in context [10]. Additionally, ferredoxin 2 (FDX2) that contains a [2Fe-2S] by itself, donates electrons (an electron per FDX2 molecule) for the cluster assembly [12].

After [2Fe-2S] cluster assembly, several proteins are involved in the delivery process, and in the synthesis of [4Fe-4S] clusters [28]. The cluster release protein complex that extracts the cluster from ISCU is formed by HSPA9 (a HSP70 protein), HSC20 (a DnaJ chaperon protein that acts facilitating the transferring from ISCU to Grx5), GRPE1 (the nucleotide exchange factor that cycles ATP on Ssq1 for cluster transfer), and Grx5 (a glutaredoxin that interacts with HSPA9 and transfers [2Fe-2S] clusters to target proteins) [28]. Even though it was previously suggested that BOLA1 might assist Grx5 in cluster delivery, new experimental evidence discarded the direct role for the heterodimer Grx5-[2Fe-2S]-BOLA1 in cluster trafficking at least in humans [29]. On the other hand, Nasta and coworkers showed that the human Grx5-BOLA3 complex can transfers [2Fe-2S]^2+^ cluster to the apo form of NFU1 yielding, in the presence of a reductant, a (4Fe-4S)^2+^ cluster [30], whereas NFU1 may transfer the latter complex to client proteins like aconitase [31]. In addition, Weiler and coworkers reconstituted the maturation of mitochondrial [4Fe-4S] aconitase, verifying the [2Fe-2S]-containing GLRX5 as cluster donor and the requirement of ISCA1/ISCA2/IBA57. FDX2 and the NADPH-dependent FDX reductase (FDXR) were also needed because electrons from FDX2 catalyze the reductive [2Fe-2S] cluster fusion [32]. The ABCB7 transporter, in the inner membrane, may be involved in the cluster and sulfur-containing compounds transport to the cytosol that are utilized for tRNA thiolation [5,6].

### 1.2. D. discoideum as A Model to Study Iron-Sulfur Cluster Assembly Physiology

The ameboid protist *D. discoideum* is commonly used in signaling studies and has become a model organism for a wide variety of biological processes and phenomena due to the following properties: (i) it has a high degree of conservation of its cellular machinery with metazoa, (ii) it can be easily manipulated genetically (it is haploid), (iii) its complete genomic sequence is available, (iv) its entire life cycle can be recreated under experimental conditions, and (v) it presents a unique life cycle, with a single-cell and a multi-cell phase that offers a wide variety of phenotypes and signaling pathways for the study [33,34,35].

Our working hypothesis is that *D. discoideum*, as an eukaryotic biological model, recapitulates the biosynthesis function of Fe-S clusters at the molecular, cellular and organism levels, making it a unique tool for the study of diseases associated with the NFS1 supercomplex and for the analysis of the effectiveness of different therapeutic strategies. We propose to use *D. discoideum* as a biological model to study the effect of various mutations associated with FRDA and ISCU myopathy at the metabolic level. In this context, this work advances the bioinformatic characterization of *D. discoideum* supercomplex and the biophysical characterization of *D. discoideum* FXN (DdFXN). We expressed the recombinant protein and studied its conformation and conformational stability. Additionally, based on sequence analysis and structural models, we studied the conservation of key residues involved in the interaction with the supercomplex and the effect of point mutation on the energetics of the FXN tertiary structure and supercomplex assembly.

## 2. Results

### 2.1. The [Fe-S] Cluster Assembly Supercomplex of D. discoideum

The examination of the genome of D. discoideum showed that the organism has the basic set of proteins that conform the supercomplex for iron–sulfur cluster assembly (Figure 1): the Cys desulfurase NFS1 (DdNFS1) precursor (XP_641773.1), the LYR motif-containing protein ISD11 (XP_635573.1); the mitochondrial acyl carrier protein ACP (XP_629874.1), a homolog of the scaffolding protein ISCU (XP_639309.1) and frataxin DdFXN (XP_629221.1). All these subunits are expected to be conserved in *D. discoideum* if the supercomplex is present and the functional mechanism is conserved. This would presumably result in a high overall identity between each homologous pair.

To analyze side by side the *D. discoideum* and the human supercomplex, first we focused on cysteine desulfurase subunits because the supercomplex consolidates its structure using this enzyme as a hub for the other proteins. In this regard, both NFS1 subunits are located at the core of the supercomplex, forming a homodimer. Each NFS1 subunit interacts with one FXN molecule, one ISCU, one ISD11, the other cysteine desulfurase subunit and the PLP cofactor [10]. As the starting point of this analysis, we located the residues from human NFS1 subunits that are close to each of the other proteins in the context of the supercomplex (less than 4 Å evaluated using the human counterpart: PDB ID: 6NZU), as well as the residues relevant for the catalytic activity of NSF1 and its interaction with PLP [7]. Next we drew a comparison between the identity corresponding to these residues and the overall identity of NFS1. Even though the results indicated that the overall percentage of identity is considerably high (59.9%, Table 1), the percentages observed for each of the NFS1 regions mentioned above are significantly higher (residues involved in NFS1–NFS1 interaction 73.2%; NFS1–ISCU interaction 77.3%; NFS1–ISD11 interaction 68.4%; NFS1–FXN interaction 80.0%; and residues involved in NFS1–PLP interaction 88.2%), which is consistent with the hypothesis of preserved interactions within the theoretical *D. discoideum* supercomplex. 

These observations form a piece of evidence in favor of the hypothesis of a shared and highly conserved Fe-S cluster assembly machinery between humans and *D. discoideum*. More specifically, the DdNFS1 sequence shows an extremely high level of conservation of the residues involved in PLP interaction. Explicitly, the following residues are conserved: Thr128, His156, Cys158, Asp232, Gln235, Ser255, His257, Lys258 and Thr295 (using the human NFS1 numbering, Appendix A). Interestingly, the active site loop that contains Cys381 (also conserved, human NFS1 numbering, Cys-loop, Appendix A), which transports the persulfide group (R-S-SH) from the NFS1 active site to the ISCU assembly site (both separated by a distance of ~20 Å) exhibits a particular feature in DdNFS1, as it contains a second Cys residue (substituted by a leucine residue in the human variant, Leu375). Whether this second Cys residue has a role in persulfide transferring is an intriguing issue that will need further research. Additionally, the stretch 272–277 of positively charged residues of NFS1, which interacts with the acidic ridge of FXN, using the human NFS1 numbering is conserved between human and *D. discoideum* Cys desulfurase enzymes.

A similar analysis can be done for ISCU. Remarkably, the structural/functional plasticity of its assembly site should be conserved among homologs, as these amino acids alternatively serve as ferrous iron binders, sulfur receptor and Fe-S cluster ligands. In this regard, DdISCU and human ISCU share the assembly site residues Cys69, Asp71, Cys95 and His137, which are involved in iron(II) binding and cluster coordination, in the human variant [12]. Furthermore, Cys138, which is also conserved, is the acceptor of the persulfide transferred by NFS1 [18,21]. In addition, the ISCU LPPVK-region (segment 131–135), which simultaneously interacts with the FXN β-sheet and the NFS1 Cys-loop, and the stretch 92–96, which interacts with NFS1, are conserved among *D. discoideum* and human variants of ISCU.

In the same fashion, the FXN regions involved in interactions with the assembly site formed by NFS1 and ISCU are highly conserved, 69% and 90%, respectively, whereas human and D. discoideum FXN sequences identity is about 30% (Table 1).

It is noteworthy that some residues from the proteins involved in pathogenesis are conserved between human and D. discoideum homologs:-Arg72 from cysteine desulfurase NFS1. It is involved in a rare disease when it is mutated to Gln (R72Q) [24].-Gly50 and Gly96 from the scaffold protein ISCU. When they are mutated to glutamic acid (G50E) and valine (G96V) result in ISCU myopathy [36,37].-Gly130, Gly137, Asn146, Gln148, Ile154, Trp155, Arg165, Trp173, Leu182, Leu186 from the kinetic activator FXN (human FXN numbering). Mutation of these residues result in FRDA.-Arg68 from tutor protein ISD11. Recent studies on human ISD11 showed that a homozygous mutation R68L is implicated in the mitochondrial genetic disorder COXPD19.

Additionally, the *D. discoideum* genome contains a complete dotation of proteins for the downstream cluster transferring process (Table 1): orthologs for HSPA9 (XP_629204.1), HSC20 (also called HSCB, or DNAJ, XP_640965.1), GRPE1 (XP_638912.1), Grx5 (XP_644145), BOLA1 (XP_644296.1) and BOLA3 (XP_639996.1); and also, the dotation of proteins for [4Fe-4S] assembly and its transferring to client proteins: the orthologs ISCA1 (XP_641284.1), ISCA2 (XP_638418.1) and NFU1 (XP_638146).

These findings as a whole suggest evolutionary constrains for the proteins that form the supercomplex for [Fe-S] cluster assembly, and more importantly, the relevance of *D. discoideum* as a model to study the physiopathology of this key metabolic pathway. This is interesting because *D. discoideum* might a useful model for drug screening [38,39]. In this context, given that FRDA is the most common among the diseases related to the supercomplex for [Fe-S] cluster assembly, we decided to start the biophysical characterization focusing on DdFXN.

### 2.2. Characterization of D. discoideum Frataxin

One of our working hypotheses is that FXN residues important for NFS1 supercomplex function should be conserved among eukaryotic FXN variants and that these residues might exhibit similar contribution to the energetics among the FXN protein homologs. To explore (*in silico*) the effect of mutation of key residues in the energetics of DdFXN, first we assembled three different tertiary structure models:(A).Based on PDB ID: 6FCO (*Chaetomium thermophilum*);(B).Based on PDB ID: 1EKG (*H. sapiens*);(C).Built using PDB ID: 2EFF (*E. coli*).

These three models have sequence identities of 44, 40 and 25%, for the mature portion of FXN, respectively. Model (A) yielded a GMQE = 0.74 (and a QMEAN = −0.79) and model (B), showed a GMQE = 0.75 (and a QMEAN = −0.64). On the other hand, model (C) yielded GMQE = 0.65 (and a QMEAN = −1.76). The GMQE score (a value from 0 to 1) reflects the expected accuracy of a model built with a given alignment, template and coverage of the target. Higher numbers indicate higher reliability. Additionally, the QMEAN-Z scores provided an estimation of the “degree of nativeness” of the structural features observed in the model on a global scale (values ~0.0 indicate good agreement between the model structure, whereas values below −4.0 are indicative of low quality). Therefore, models (A) and (B) based on *C. thermophilum* and human FXN, respectively, seem to have similar quality (Figure 2). The main structural difference between these models is located at the C-terminal helix and C-terminal region. The 6FCO based model has a two-turns shorter C-terminal helix and the last residues are not modeled. Some smaller differences are present in loop 1, the stretch between helix α1 and strand β1. On the other hand, the beta sheet is structurally conserved (Appendix A).

Even though EcFXN exhibits lower sequence identity with DdFXN than the human or *C. thermophilum* variants, the *E. coli* FXN template also seems to be useful to capture the relevant features of the FXN three-dimensional structure, more likely as a consequence of the fold conservation between bacteria and eukaryotes (data not shown).

More importantly, the analysis of secondary structure predictions using JPRED 4 [40] (http://www.compbio.dundee.ac.uk/jpred/) suggested that the C-terminal α-helix may involve residues Thr173-Pro174 to Leu186-Cys187 (Appendix A). This result is in agreement with models (B) and (C) but not with model (A), which exhibits a shorter α-helix finishing at positions Ile183-Asn184.

Molecular dynamics simulation results (Appendix A) shows that the model generated from the human template of FXN, model (B), remains relatively stable, although the root of the mean squared deviation (RMSD) is relatively high compared to previous MD simulations of wild-type human FXN [41]. This increase in RMSD may occur due to a reorganization of the secondary structure, particularly in the N-terminal α-helix, where an internal turn of the structural element is partially lost. We plan to continue analyzing the temperature dependence of the dynamics using this model, while at the same time we will attempt to crystallize and elucidate the X-Ray structure to validate the results presented here.

Figure 2 show a superposition between the human FXN and the DdFXN model built using the former as the template (PDB ID: 1EKG). It is worthy of mention that ten of the conserved residues between both frataxin variants correspond to positions related to FRDA in humans. Six of these residues are located in the β-sheet surface (Figure 2B) and, according to the structure of the cysteine desulfurase NFS1 supercomplex (PDB ID: 6NZU [10]), these residues are involved in protein–protein interactions, with ISCU and also with the catalytic loop of NFS1 and the C-terminal region of NFS1. Thus, it is expected that the alteration of this cluster of residues results in the destabilization of the supercomplex in D. discoideum. Interestingly, the other four residues that are conserved between the *D. discoideum* and humans protein and are related to FRDA are located in the core of the protein (Figure 2C), and their mutation is predicted as destabilizing of the FXN structure.

We wondered about the effect of mutations on the energetics of DdFXN structure. To evaluate whether the mutations found in FRDA-related variants of human FXN have similar effects on the energetics of the DdFXN structure, we simulated mutations by using the FOLDX software [43,44]. Figure 3 presents the results corresponding to the computational alanine scanning mutagenesis for all residues of human and DdFXN. Remarkably, there is a good correlation between the estimated effect on the conformational stability of DdFXN ($\Delta \Delta G_{\mathrm{NU},\;\mathrm{DdFXN}}^{o})$ and that estimated for the stability of the human FXN structure ($\Delta \Delta G_{\mathrm{NU},\;\mathrm{hsFXN}}^{o}).$
Figure 3A shows the correlation for all residues. Non conserved residues, as expected, showed a weaker correlation (Figure 3B), whereas good correlations were observed for the conserved residues (Figure 3C). In particular, the FRDA related-residues (Gly130, Gly137, Asn146, Gln148, Ile154, Trp155, Arg165, Trp173, Leu182, Leu186 in the human variant of FXN) and Gln153 (we included the already studied variant Q153A [42,45]) and Gly122, Gly129, Asn137, Gln139, Ile145, Trp146 and Arg156, Trp166, Leu175, and Leu179 in DdFXN, and Gln144) exhibited a good correlation (Figure 3D). Given that these residues form clusters of contacts among themselves (one cluster in the core and a second one on the β-sheet surface, e.g., the π-cation pair interaction established between Trp155 and Arg165), we deduce that the conservation of these environments is manifested as energetic constrains. It is worthy of note that the poorest correlation among the FRDA set of residues was observed for Asn146. The rotamers of this residue and of Gln148 seem to be energetically coupled in the case of the human variant, whereas this would not occur in the case of DdFXN.

### 2.3. Protein Expression

The mature form of DdFXN was expressed in *E. coli* as a soluble protein with a high yield. Purification included two parts, first an ionic exchange step (DEAE) and after that a preparative size exclusion chromatography (G100). This protocol guaranteed more than 100 mg of protein per L of culture (>98%). The purified protein exhibited approximately two free thiols per protein molecule (1.8 ± 0.2). The cysteine residues would be at a distance no longer than 6–8 Å (between sulfur atoms, Swiss model). Thus, a small conformational change would allow the oxidation of these Cys residues to form intramolecular or intermolecular disulfide bonds. The preservation of the protein at −70 °C impeded the oxidation of the cysteine residues.

To evaluate the oligomerization state of DdFXN, we ran a size exclusion chromatography (SEC)-HPLC experiment under native conditions. The protein was loaded in a Superose 6 column and its absorbance was monitored at 280 nm. The result indicated that DdFXN behaves in solution as a monomer. The inferred molecular mass, by means of a calibration curve with globular proteins and molecular weight standards, is 16.0 ± 0.2 kDa (Figure 4), in reasonable agreement with its theoretical mass 13.3 kDa. It is worthy of mention that acidic proteins (low pI) usually are excluded from the Superose matrix because of the negatively charged surface and their unfavorable interaction with the matrix.

Similar results were observed when the protein was preserved at −70 °C and when the protein was incubated for 30 days at 4 °C suggesting a very low tendency to aggregate. Remarkably, DdFXN does not form visible aggregates, even at high concentration (25 mg·mL^−1^, in buffer 20 mM Tris-HCl, pH 7.0). DdFXN was subjected to robotic crystallization screening using the sitting-drop vapor diffusion method. Even though we were not able to find crystals after a month (plates were monitored weekly), is important to mention, as a preliminary experimental evidence, that the protein formed visible aggregates first in drops under acidic conditions (Figure 5).

### 2.4. Spectroscopic Characterization of DdFXN

Some structural features of DdFXN could be inferred from a simple analysis of UV absorption and from the 4th derivative spectra that highlight sharp bands present in the former. Figure 6 shows the spectra corresponding to the combination of N-acetyltryptophanamide (NATA), N-acetyltyrosinamide (NAYA) and free phenylalanine (Phe) in solution as present in DdFXN (three Trp, three Tyr and five Phe) and the corresponding spectra of purified DdFXN. The analysis of the spectra indicated that the Trp residues are located in an apolar environment, as judged by the red shift observed: a band at 293.3 nm for the Trp in the context of protein compared to the 189.0 nm band corresponding to the NATA, for the combination NATA, NAYA and Phe. This is specific of a folded structure. Similar results were obtained after freeze (at −70 °C) and thaw.

To evaluate the secondary structure content of recombinant DdFXN we monitored its CD signal in the far-UV region (Figure 7A). The shape of the CD spectra (Table 2) was compatible with an α/β protein and the predictions of secondary structure content using the Bestsel software were compatible with that observed for the model of the DdFXN structure.

Additionally, we studied the tertiary structure of the protein by means of near-UV CD spectroscopy and by tryptophan fluorescence. The former to evaluate the presence of rigid asymmetric environments in the vicinity of the aromatic residues and the latter to detect shifts in the tryptophan emission wavelength that occur depending on the chemical environment of this side-chain (emission occurs at 350 nm when tryptophan is in a polar environment, while it is blue shifted when it occurs from an apolar media like the core of the protein). The results showed that DdFXN is well folded, exhibiting rigid tertiary structure and a hydrophobic core (Figure 7B,C).

A noteworthy feature of the near-UV CD spectra is the very sharp and positive band at 290.7 nm, corresponding to tryptophan residues. This band is completely obliterated when the protein is incubated in the presence of 5 M of urea. More experiments will be carried out to establish if this band is a product of the tryptophan-tryptophan motif (-W-W-) present in this protein or, on the other hand, if this band is the result of one of the tryptophan residues alone.

### 2.5. Conformational Stability of DdFXN

The differences observed in the tryptophan fluorescence spectra of DdFXN in the presence or in the absence of high urea concentrations (Figure 7C) indicated that this probe may be adequate to be used for conformational stability measurements by monitoring the tertiary structure content in the protein sample. Isothermal (20 °C) urea-induced equilibrium unfolding experiments were carried out (Figure 8A,B). The fitting of a two-state model (N⇄U) suggests that DdFXN is unstable ($\Delta G_{\mathrm{NU}}^{o}=4.3\;\mathrm{kcal}\cdot {\mathrm{mol}}^{-1}$) compared to the human variant ($\Delta G_{NU}^{o}=\;9.0\;kcal\cdot mol^{-1}$ [49,50,51]), and the presence of a shorter C-terminal region (CTR) in the former predicts this feature (as in the case of *E. coli* and yeast variants, [52,53]). On the other hand, the protein unfolds in a cooperative fashion with a $m_{\mathrm{NU}}=\;1.55\;\mathrm{kcal}\cdot {\mathrm{mol}}^{-1}\cdot M^{-1}$, very similar to that predicted for the complete unfolding of a 116-residue protein ($m_{\mathrm{NU}}=\;1.46\;\mathrm{kcal}\cdot {\mathrm{mol}}^{-1\cdot }M^{-1}$), using the correlation given by Myers and coworkers [54].

Additionally, we investigated the reversibility of the temperature-induced unfolding process, monitored by CD in the far-UV region. The results showed that unfolding is nearly fully reversible, as judged by the recovery of the signal upon cooling the solution after the unfolding ramp (Figure 8C). The fitting of a two-state model to the data using a $\Delta C_{P}=1.6\;\mathrm{kcal}\cdot {\mathrm{mol}}^{-1}\cdot K^{-1}$, which is compatible with a protein of 116 residues, yielded a lower value of stability ($G_{\mathrm{NU}}^{o}=2.7{\;\mathrm{kcal}\;\mathrm{mol}}^{-1}$) compared to that obtained in urea-induced unfolding experiments. This result might suggest that the temperature induced unfolding is not complete and that the unfolded state might have a residual structure. Lower $\Delta C_{P}$ values resulted in an increase in the difference in free energy (Appendix A).

### 2.6. The Evolutionary Tree of Frataxin and DdFXN: Function and Stability Related Features

To study the signatures shared by the amino acid sequences from DdFXN and the human frataxin in an evolutionary context, we situated them in the tree of life. Firstly, we assessed the spread of FXN homologues across the different groups of organisms and then we selected FXN homologues from 54 genomes representing them. As firstly noted by Gibson and collaborators in 1996 [55], FXN is present in Eubacteria, only in proteobacteria, and absent in Archaea. Although at that time there were not as many genomes available, the present scenario is virtually the same scenario (Appendix A); only a few exceptions have appeared; some may be the result of horizontal transfer events, or simply annotation errors due to contamination in sequencing projects. Nevertheless, as Gibson suggested, FXN was probably present in the proto-mitochondrial endosymbiotic proteobacterium. It is worth noting that computational algorithms used for protein domain prediction, such as Pfam hmm (http://pfam.xfam.org/) cluster within the same domain ID both the eukaryotic and bacterial FXN homologues. As expected, FXN eukaryotic homologues present an extended N-terminus containing the mitochondrial targeting signal (MTS). Since MTS is highly divergent, we removed the N-terminal part of the sequence prior alignments (see Material and Methods); the alignment was manually curated removing positions where most sequences had gaps and cutting all positions that extend beyond the last amino acid of the human FXN rendering an alignment of 120 positions. The conservation of FXN along evolution is evident when all sequences are aligned (Figure 9, Appendix A: All54G_cured_alignment.aln).

Out of 120 positions in the alignment, 49 positions (40.8%) were conserved in more than 50% of the genomes analyzed, 21 positions (17.5%) in more than 80% and 10 positions (8.33%) in more than 90%; three positions were absolutely conserved, one glycine and two tryptophan residues (Gly162, Trp155 and Trp173 in human). Although conservation is high along the frataxin domain, it is outstanding in the central part of the alignment in a region that contains more than 40 highly conserved amino acids that contains the Motif 1 described by Gibson et al. [55], which includes Trp155 and Gly162. The second half of the alignment shows a region with some high conserved positions, such as the second absolute conserved Trp (Trp173), Leu182, Leu186, Glu189; the latter conserved in eukaryotic genomes and only in *Rickettsia prowazekii* among the bacterial genomes included. The end of the alignment, that was set to end with the human FXN, is less conserved and some of the protein did not extend that much. Still the degree of conservation is such that ML-phylogeny could not resolve most of the tree branches for several groups of organisms (Appendix A).

Given that some of the FRDA mutations found in human FXN resulted in a functional deficit and others resulted in a significant destabilization of the conformation, and the consequent decrease in FXN concentration inside the mitochondria, we assessed these positions of FXN sequences among representative organisms through the diverse evolutionary tree branches (Figure 10).

As we mentioned above, one of the most astounding results is that Trp155 (human numbering), which is partially accessible to the solvent (with a solvent accessible surface area (SASA) = 29%), is completely conserved. All organisms included, without exceptions, exhibit a Trp residue at this position. Why this residue is so conserved? If one inspects the structure of the human supercomplex (NFS1/ACP-ISD11/ISCU/FXN)_2_ in detail, it can be observed that this residue is not only near the docking surface of ISCU, but also it is at Van der Waals distance of the [2Fe-2S] assembly site of the latter and at Van der Waals distance from Leu386 from NFS1 (in fact the SASA corresponding to Trp155 drops to 5% when FXN is docked to the supercomplex). Therefore, Trp155 may be a key residue for persulfide group transferring from NFS1 Cys loop to ISCU assembly site and cluster assembly. The analysis of the FXN–ISCU interface (PDB ID: 6NZU, [10]) shows that the invariant Trp155 may establish π–π interactions with His137 from the ISCU assembly site. This interaction might finely control the accessibility and dynamics of Cys138, which is the receptor for the persulfide, exerting (or at least contributing to) the kinetic activation. Thus, Trp155 may have a dual role, establishing protein–protein interactions with ISCU consolidating the structure of the supercomplex and in the fine tuning of the enzymatic catalysis. Interestingly Asn146 from FXN (human numbering), which is a Van der Waals distance of Trp155 and mutated to Lys in FRDA patients, is also highly conserved, all the eukaryotic frataxins, with the exception of Naegleria and Trichomonas that present Ser, contains an Asn residue at this position, whereas Bacteria show Asn/Ser (polar non charged small side chain). In the context of the supercomplex, Asn146 is also at Van der Waals distance from ISCU His137 (the histidine from the ISCU assembly site) and Leu386 from NFS1. Residue Gln148 from FXN is also interesting. In the supercomplex, it is at Van der Waals distance from ISCU Cys69 (assembly site). Therefore, it is more likely to be conserved for protein function. It is always Gln with the exceptions of Bacteria (Gln or His), Toxoplasma and Jakobida (His in both cases).

On the other hand, Trp173 is also absolutely conserved. All the sequences contain a Trp at this position. It is worthy of mention that Trp173 contributes to the stability of the human FXN in an extreme fashion, non-published results of our laboratory indicated that this is a key residue and the FRDA W173G variant results in a high destabilization of the FXN conformation (in the human FXN, Trp173 has a SASA = 0%). To a lower extent but also conserved (always an apolar side chain), Leu186 is a core position (SASA = 0% in human FXN) clearly related to the conformational stability of FXN. Not all FRDA residues are conserved. In fact, for three of the FRDA positions we can even find the amino acid residue corresponding to the FRDA mutation, as the residue present in other organisms (Appendix A: matrix_all54G.xlsx); first, the G137V substitution is present in FXN from *Trypanosoma cruzi*; second, the L182F substitution appears in Naegleria, some bacteria and some fungi; the last case is quite puzzling, the H183R is conserved in all Dictyostelids but not in the other Amoebozoan genome analyzed (*Acanthamoeba castellanii*) which is the only non-metazoan that presents a histidine in this position.

## 3. Discussion

Approximately 2% of the human genes encode iron-proteins. From this set, about 17% of the proteins contains iron–sulfur [Fe-S] clusters [57], which are essential groups involved in electron transfer and enzymatic reactions, in several biochemical pathways. Among them, NADH dehydrogenase, Rieske protein (from complexes I and III of the respiratory chain, respectively), and aconitase and succinate dehydrogenase enzymes (from Krebs cycle) contain iron–sulfur clusters. These chemical groups are also involved in sensing DNA breaks, DNA replication and repairing [58].

In humans, alterations in 62% of the mitochondrial Fe-S cluster containing proteins were associated to pathologies [57]. In eukaryotic cells, [2Fe-2S] cluster assembly in the mitochondria is activated by FXN and the decrease in FXN functionality or expressions results in Friedreich’s Ataxia. In this study, as the first part of a comprehensive project that involves the full characterization of Fe-S cluster biosynthesis in *D. discoideum*, the conformation and stability of DdFXN was experimentally investigated and placed in context by using a battery of bioinformatic tools, taking into account the proteins involved, the presence of the *D. discoideum* homologs in the genome of this organism and the structure of the supercomplex for cluster assembly.

DdFXN and the human variant exhibit ~40% of amino acid residue identity (mature region). In particular, several of the conserved residues are involved in protein–protein interactions and are important for the function of the human supercomplex for iron–sulfur cluster assembly, as judged by studies carried out for the human counterpart. In addition, some conserved residues have been proved by previous research to be crucial for the conformational stability of the human variant.

Whether similar mutations in the DdFXN and the human variant have similar effects on the biological function of this proteins is an intriguing issue because these proteins differ in their conformational stability and also the organisms differ regarding the range of temperature in which they live. In fact, DdFXN is drastically more unstable than the human FXN [49], and *D. discoideum* lives at 20 °C, a value significantly lower than our 37 °C. Moreover, the temperature in full active mitochondria may be significantly different from these values. In fact, in the case of mammalian cells this value might increase to 50 °C [59,60,61]. Further research is necessary to evaluate the temperature inside the mitochondria of *D. discoideum.*

As we have already mentioned, the lower stability of DdFXN is correlated with the presence of a shorter C-terminal region (CTR) [53]. For the human variant, it was previously shown that apolar residues located in the CTR specifically stabilize the native state without altering the stability of the transition states or that of the unfolded state [51]. The counterpart for some of these residues is still present in the shorter CTR of DdFXN: Met191 and Ile193, which correspond to Leu198 and Leu200, respectively, in the human variant (this inference is also validated by a global multiple sequence alignment (MSA), Appendix A). Taking this information into account, one might reason that Met191 and Ile193 might contribute to the stability of DdFXN in a similar fashion as the corresponding positions do to the human FXN. In the latter, the L198A and L200A mutations decrease the conformational stability in 2.5 and 3.0 kcal mol^−1^, respectively [51], which amount to 5.5 kcal mol^−1^, exceeding by far the amount corresponding to the global stability of the DdFXN determined in the same experimental conditions. Therefore, we inferred that the CTR is a key element for the stability of the protein not only in the human variant but also in DdFXN. The prediction made by means of the FOLDX program indicated that the destabilization effects of the M191A and I193A mutations are 2.7 and 1.9 kcal mol^−1^, respectively (a total of 4.6 kcal mol^−1^), suggesting that the inferences regarding the role of these residues and the CTR are most likely correct.

The fact that at least four FRDA-related residues located in the core of FXN are conserved between DdFXN and the human variant (Ile154, Trp173, Leu182 and Leu186 in the human variant and Ile145, Trp166, Leu175 and Leu179 in DdFXN), suggests that the mutation of these residues may be also deleterious in *D. discoideum*. They might destabilize the DdFXN structure to a higher extent.

As a whole, the analysis of stability and conservation suggests similar modes to stabilize the tertiary structure of both FXN variants. In this context, a decrease in the stability of DdFXN may reduce the levels of FXN concentration, as occurs in the case of some human variants. In this regard, it was shown that the mutation G137V destabilizes the recombinant protein without altering the activity of the protein (activation capability) causing of FRDA because of the low FXN concentration inside the mitochondria [62]. Therefore, the G137V variant will be a good candidate for further studying the effect of altering stability in the cell molecular physiopathology in *D. discoideum.*

DdFXN and other FXN variants share the characteristic acidic ridge involving the N-terminal α-helix, loop 1 and strand β1. In the case of DdFXN, it is formed by thirteen Glu and Asp residues. It is worthy of mention that in the human variant this region of the protein interacts with a positively charged stretch of residues of NFS1. On the other hand, for several variants, including the human FXN, it was observed that this negatively charged region can interact with iron(III) and iron(II), with *K*_D_ in the 10–50 μM range [19,63,64]. For this reason, FXN was first postulated as an iron chaperone. The mutation of acidic residues by alanine in the case of the yeast variant resulted in a significant increase in the stability of the protein [65], suggesting that the conservation of the acidic ridge is related to the protein function. Whether or not DdFXN is also involved in iron homeostasis and metal ion bioavailability inside the mitochondria, is a specific issue that might be experimentally evaluated. Since essential features of the supercomplex are conserved, and *D. discoideum* model presents several advantages (among them, the amoeba is haploid, numerous molecular tools were developed for its study [66,67], and iron homeostasis is important for *D. discoideum* growth and survival [68,69], we expect to obtain some general answers on frataxin roles, making possible the functional dissection based on complementary mutations carried out on the supercomplex subunit pairs.

The inspection of the *D. discoideum* genome showed that it includes the basic set of genes for Fe-S cluster assembly. The computational analysis of the *D. discoideum* and the human subunits also suggest a highly conserved Fe-S cluster assembly machinery. Functionally relevant regions of the supercomplex show an extremely high level of residue conservation. Nevertheless, intriguing issues such as a second cysteine residue in NFS1 Cys loop, need further research.

We think that the study of the evolutionary constrains for the proteins that form the supercomplex will be relevant to the study of the physiopathology of this key metabolic pathway. In this regard, it will be very important to assess the supercomplex function by a mutational approach, including those mutations in FXN, NFS1, ISCU and ISD11 that determine severe human diseases.

As the *D. discoideum* genome contains a complete dotation of proteins for the downstream cluster transferring process it will be also important to evaluate the specificity of these proteins and the degree of coevolution and exchangeability of the human and *D. discoideum* subunits.

## 4. Materials and Methods

### 4.1. Computational Analysis of the Protein Structure

Protein visualization, evaluation of its secondary structure content, and overall analysis of the protein structure (calculation of interatomic distances, etc.) were done with Yasara [46]. Figures of molecular structures were prepared with the same program.

By means of Swiss-model (https://swissmodel.expasy.org/) [70], the following templates were used to build models: (i) frataxin-like protein from *Chaetomium thermophilum* (PDB ID: 6FCO, [71]) and (ii) the human variant of frataxin (PDB ID: 1EKG, [72]). Calculations of the energetics were performed using FOLDX (under Yasara [43,44]). First models and X-ray structures were submitted to the repair routine. After that, in silico mutations were carried out by applying the mutate residue routine. Finally, stability of the object and sequence detail routines were performed.

### 4.2. Molecular Dynamics Simulations

The coordinates correspond to the homology model generated using human FXN as a template (PDB ID: 1EKG) solvated in an octahedral TIP3P water molecules box. Protonation states of amino acid residues were set to correspond to those at pH 7.0. A standard minimization, thermalization and equilibration protocol was applied. The temperature was held constant at 290 K and constant pressure at 1 atm using the Berendsen thermostat and barostat. The SHAKE algorithm was applied to all bonds that involved hydrogen atoms. A production run of 300 ns was performed using Amber18 [73] using the pmemd.cuda module and the ff14SB force field [74].

### 4.3. Protein Expression and Purification

The DNA fragment coding for the mature form of *D. discoideum* frataxin (DdFXN) optimized for *Escherichia coli* protein expression was synthetized by BIO-BASIC Inc. (To, Canada). The cDNA product was cloned into the pET9a plasmid vector (kanamycin ^R^) and sequenced. *E. coli* BL21 (DE3) bacteria cultures were grown at 37 °C in 2 L Luria–Bertani (LB) medium with agitation (280 rpm). Protein expression was induced at OD_600nm_ = 1.0 by addition of 1.0 mM isopropyl-β-d-thiogalactopyranoside. After induction for 4 h at 37 °C and 280 rpm, the culture was centrifuged at 6000 rpm and the pellet was stored at −20 °C. Bacteria were resuspended in 25 mL of lysis buffer (20 mM Tris-HCl, 10 mM EDTA, pH 7.0) and cells were disrupted by sonication in an ice-water bath, followed by centrifugation at 10,000 rpm for 30 min at 4 °C. For the purification of mature DdFXN (116 residues, pI 4.4), the soluble fraction was carefully loaded onto a DEAE ion exchange chromatography column (DE52 matrix) and eluted with a 200-mL linear gradient (0.0—1.0 M NaCl) in buffer 20 mM Tris-HCl, 1 mM EDTA, pH 7.0. Subsequently, fractions with DdFXN (identified by SDS-PAGE) and low DNA concentrations (analyzed using a V730 BIO spectrophotometer, JASCO, Japan) were subjected to size exclusion chromatography on a preparative Sephadex G–100 column (93 cm, 62.7 cm), previously equilibrated with buffer 20 mM Tris-HCl, 100 mM NaCl, 1 mM EDTA, pH 7.0. This protocol yielded >98% pure DdFXN as confirmed by mass spectrometry (ESI-MS) (theoretical molecular mass value: 13,326.9 Da, considering the N-terminal methionine residue). The protein concentration was determined by UV spectrophotometry using the extinction coefficient ε_280nm_ = 20,970 M^−1^ cm^−1^ calculated from the amino acid sequence.

### 4.4. Thiol Quantification

The chemical reactivity toward the Ellman reagent (DTNB, Sigma-Aldrich. St. Louis, MO, USA) was determined at 20 °C in a buffer solution containing 20 mM Tris-HCl, 100 mM NaCl, pH 8.0. A 2.0 mg mL^−1^ DTNB solution (5 mM) was prepared in a 100 mM sodium phosphate buffer, 1 mM EDTA, pH 7.0. The protein was used at a concentration of 30 μM and 5 μL of DTNB stock was added to begin the reaction in a final volume of 500 μL. The absorbance at 425 nm was monitored during 10 min in a 1-cm pass length thermostated cell. A blank without protein was collected to subtract the absorbance of the reaction mixture (without any thiol group) at 425 nm.

### 4.5. Protein Aggregation and the Evaluation of Crystallization Conditions

DdFXN was subjected to robotic crystallization screening at 2.5 and 5.0 mg mL^−1^ using 96-well sitting-drop vapor diffusion Greiner 609120 plates (Monroe, NC, USA) and a Honeybee963 automatic dispenser (Digilab, Marlborough, MA, USA). A total of 288 different conditions from Jena Bioscience commercial kits (Jena, Germany) were screened. Every droplet consisted of a 1:1 mixture of protein (in 10 mM Tris-HCl, pH 7.0) and crystallization solution in a total volume of 700 µL. Plates were stored at room temperature and visualized every week for a total of 80 days on a cold light source Olympus SZX16 stereomicroscope (Tokyo, Japan).

### 4.6. Size Exclusion Chromatography

SEC-HPLC was performed using a Superose-6 column (GE Healthcare, Chicago, IL, USA). Protein concentration was 40 μM, a volume of 50 μL was typically injected, and running buffer was 20 mM Tris-HCl, 100 mM NaCl, 1 mM EDTA, pH 7.0. The experiment was carried out at room temperature (∼25 °C) at a 0.5 mL·min^−1^ flow rate. A JASCO HPLC instrument equipped with an automatic injector, a quaternary pump and a UV-VIS UV-2075 detector was used. The elution was monitored at 280 nm.

### 4.7. Characterization by UV 4th Derivative Absorption Spectra Analysis

Absorption spectra (240–340 nm range, using a 0.1-nm sampling interval) were acquired at 20 °C with a V730 BIO spectrophotometer (JASCO, Japan). Ten spectra for each sample were averaged, and blank spectra (averaged) subtracted. A smoothing routine was applied to the data by using a Savitzky–Golay filter and after that, the 4th derivative spectra were calculated [75,76].

### 4.8. Characterization by Circular Dichroism Spectroscopy

CD spectra measurements were carried out at 20 °C with a Jasco J-815 spectropolarimeter calibrated with (+) 10-camphor sulfonic acid. Far and near-UV CD spectra were collected using cells with path lengths of 0.1 and 1.0 cm, respectively. Data were acquired at a scan speed of 20 nm min^−1^ and five scans were averaged. Finally, blank (buffer) scans were averaged and subtracted from the spectra. The buffer solutions used were 20 mM Tris-HCl, 100 mM NaCl, pH 7.0 (near-UV) and 2 mM Tris-HCl, 10 mM NaCl, pH 7.0 (far-UV, to increase the signal-to-noise ratio in the 190–200 nm region). The protein concentrations were 30 µM and 8 µM for near-UV and far-UV, respectively. The calculation of the secondary structure content based on the CD spectra was performed by using the program Bestsel (http://bestsel.elte.hu/index) using two different data ranges (190–250 nm or 200–250 nm) [47,48].

### 4.9. Characterization by Fluorescence Spectroscopy

Steady-state fluorescence measurements were performed in an Aminco-Bowman Series 2 spectrofluorometer equipped with a thermostated cell holder connected to a circulating water bath set at 20 °C. A 1.0-cm path length cell was used. When the intrinsic fluorescence of proteins was measured, the excitation wavelength was set to 295 nm and emission data were collected in the range 310–450 nm. The spectral slit-width was set to 3 nm for both monochromators. The protein concentration was 3.0 µM and the measurements were done in buffer 20 mM Tris-HCl, 100 mM NaCl, pH 7.0.

### 4.10. Conformational Stability

Isothermal unfolding experiments were carried out by incubating DdFXN with 0–7.2 M urea in a buffer solution consisting of 20 mM Tris-HCl, 100 mM NaCl, pH 7.0 for 3 h at 20 °C. The process was followed by tryptophan fluorescence. For the calculation of the thermodynamic parameters, a two-state unfolding mechanism was assumed, as only native (N) and unfolded (U) conformations exist at equilibrium [54]. Additionally, unfolding transitions as a function of temperature were monitored by the CD at 220 nm. Experiments were carried out in 20 mM Tris-HCl, 100 mM NaCl, pH 7.0. The protein concentration was 8.0 µM and a cell with a path length of 1.0 cm was used. The temperature was increased from 20 to 90 °C at a constant rate of 1 °C min^−1^, sampling at intervals of 0.1 °C. To extract the thermodynamic parameters, a two-state model was fitted to the data as mentioned in previous works [49,77].

### 4.11. Iron-Sulfur Cluster Biogenesis Orthologs in D. discoideum and Frataxin Sequence Analysis

All human proteins involved in basic machinery of iron–sulfur Clusters were downloaded from Uniprot based on [28]. Reciprocal best hit was performed with *D. discoideum* proteome (NCBI and Dictybase [78,79]) using blastp. Percentage sequence identity, coverage, e-value, functional domains and several other features were extracted (Table 1, Figure 1 and Appendix A). To get a comprehensive catalogue of FXN homologues we implemented a combined approach: (i) we made blastp (https://blast.ncbi.nlm.nih.gov/Blast.cgi) using human frataxin (NP_000135) as query using default parameters and changing Max Target Sequences to 5000; we then filtered the results using 0.01 as cut off for e-value; (ii), on the other hand, we searched the Pfam database (Pfam 33.1, http://pfam.xfam.org/family/PF01491#tabview=tab7) for phylogenetic distribution of protein carrying a frataxin_cyay (PF01491) domain. We compared the phylogenetic distribution in both set of data (Appendix A) and selected FXN homologues from 54 genomes (−2).

Prior to alignment the N-terminus of each sequence was trimmed up to the first amino acid included in the PF01491 domain. The sequences where then aligned using MAFFT v.7453 [80]. Visualization and manual curation of alignment was carried out using AliView software (v1.26) [81]. We performed maximum-likelihood (ML) phylogeny reconstructions using MEGAX (v 10.1.7) [82], bootstrap method with 500 replications, JTT model for amino acid substitution, assuming uniform rates. Graphical LOGO representations of multiple alignments were performed using WebLogo with default settings [83].

## Figures and Tables

**Figure 1 ijms-21-06821-f001:**
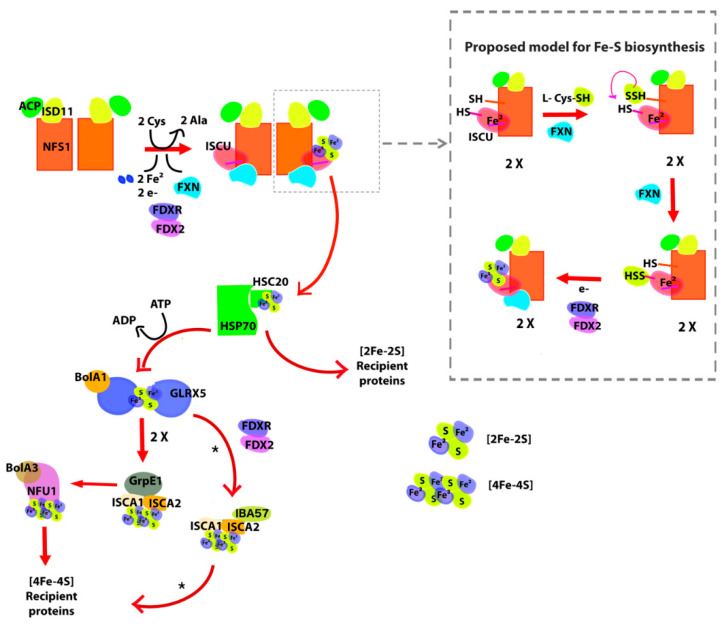
General scheme of iron sulfur cluster biogenesis in mammalian cells. Briefly, the ISU (in this case, (NFS1/ACP-ISD11/ISCU/FXN)_2_ supercomplex (where ACP is the acyl carrier protein, ISCU is the iron–sulfur cluster assembly enzyme and FXN is frataxin) synthesizes [2Fe-2S] clusters. The ISCU protein functions as a scaffold and requires a persulfide given by NFS1. FXN accelerates this reaction. The persulfide of ISCU is then reduced into sulfide by FDX2, leading to the formation of [2Fe-2S] clusters. The [2Fe-2S] structure is then transferred to the HSP70-HSC20 chaperone system and then to recipient proteins through the HSP70-HSC20 chaperone system, to GLRX5-BolA1 system with coupled ATP hydrolysis or take part in sulfur compounds that exit the mitochondria. If the cluster goes through GLRX5, this protein dimerizes and the cluster is transferred to the ISA complex (ISCA1-ISCA2-GrpE1) that lead the formation of [4Fe-4S] clusters. Finally, the [4Fe-4S] cluster binds to NFU1-BolA3 proteins and is transferred to [4Fe-4S] recipient proteins. Asterisk marks that other pathways for [4Fe-4S] assembly and release have been reported [32].

**Figure 2 ijms-21-06821-f002:**
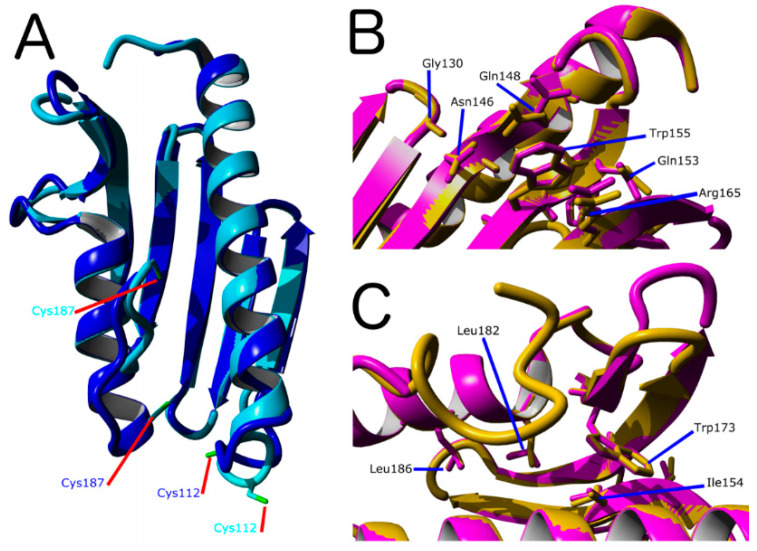
Molecular Model for of *D. discoideum* frataxin protein (DdFXN). (**A**) Structural alignment of models made using PDB IDs 1EKG (blue) and 6FCO (cyan) as templates. As a reference, residues Cys112 and Cys187 are shown. (**B**) Superimposition of the DdFXN model (magenta) and the human FXN variant (1EKG, orange) indicating the conserved residues among the variants involved in Friedreich’s Ataxia (FRDA), residues located in the β-sheet surface in human variant (Gly130, Gly137, Asn146, Gln148, Trp155, Arg165) and in DdFXN (Gly122, Asn137, Gln139, Trp146, and Arg156), and (**C**) residues located in the core of the protein in the human variant (Ile154, Trp173, Leu182 and Leu186) and in DdFXN (Ile145, Trp166, Leu175 and Leu179). Structural alignments were performed using Yasara. In addition, we included Gln153 (in the human variant) and Gln144 (in the DdFXN) given that mutation Q153A was reported to have a large effect on frataxin function [42].

**Figure 3 ijms-21-06821-f003:**
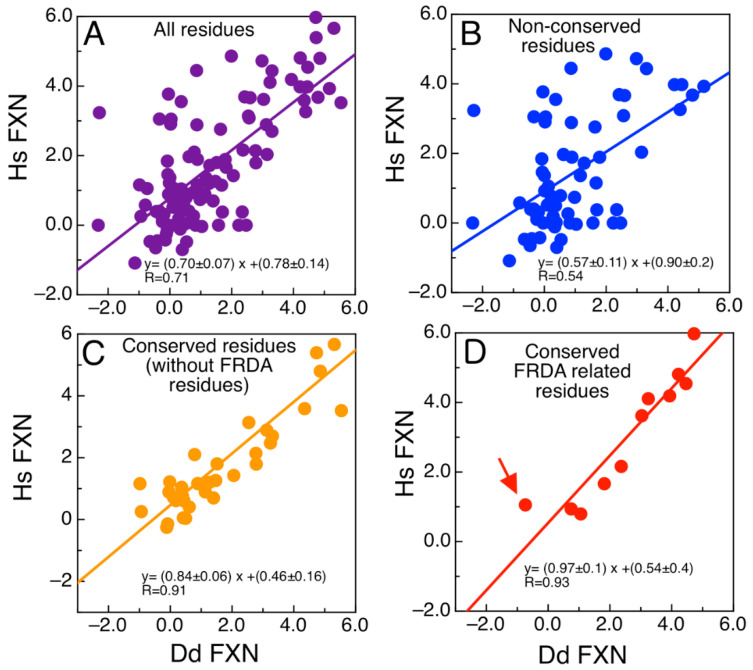
Analysis of the Energetics of the DdFXN Structure Model. Correlation of free energy differences between the alanine scanning using FOLDX corresponding to the DdFXN ($\Delta \Delta G_{\mathrm{NU},\;\mathrm{DdFXN}}^{o})$ and human variant ($\Delta \Delta G_{\mathrm{NU},\;\mathrm{hsFXN}}^{o})$ for (**A**) all the residues, (**B**) only residues that are not conserved, (**C**) only residues that are conserved but without including FRDA-related residues and (**D**) showing only residues conserved and FRDA-related. The red arrow indicates the N146A mutation. Free energy differences for each Ala mutant was calculated as follows: e.g., for N146A variant the value of $\Delta \Delta G_{\mathrm{NU},\;\mathrm{hsFXN}\;}^{o}=\Delta G_{\mathrm{NU},\;\mathrm{hsFXN}\;N146K}^{o}-\Delta G_{\mathrm{NU},\;\mathrm{hsFXN}\;\mathrm{wt}}^{o}$. DdFXN was modelled using the human (PDB ID: 1EKG) variant as template.

**Figure 4 ijms-21-06821-f004:**
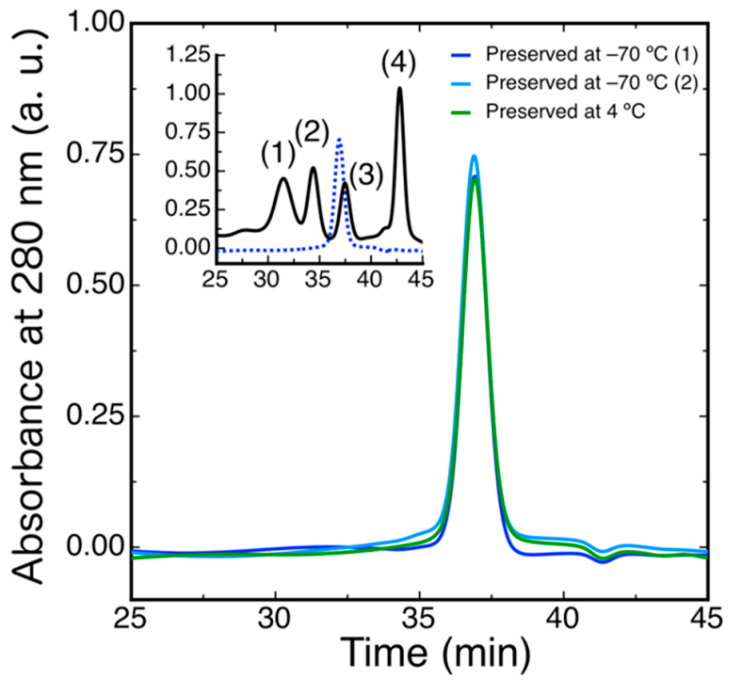
Hydrodynamic Characterization of DdFXN by means of SEC-HPLC. DdFXN (40 µM, 50 µL) was loaded in a Superose 6 column equilibrated in buffer 20 mM Tris-HCl, 100 mM NaCl, 1 mM EDTA, pH 7.0. Sample maintained during 30 days at 4 °C (green line) and samples preserved at −70 °C (blue and cyan lines). The inset shows the chromatogram corresponding to the molecular mass markers (black line): (1) gammaglobulin (158 kDa), (2) ovoalbumin (44 kDa), (3) myoglobin (17 kDa), and (4) vitamin B12 (1350 Da) and a chromatogram corresponding to DdFXN, for reference (blue dashed line).

**Figure 5 ijms-21-06821-f005:**
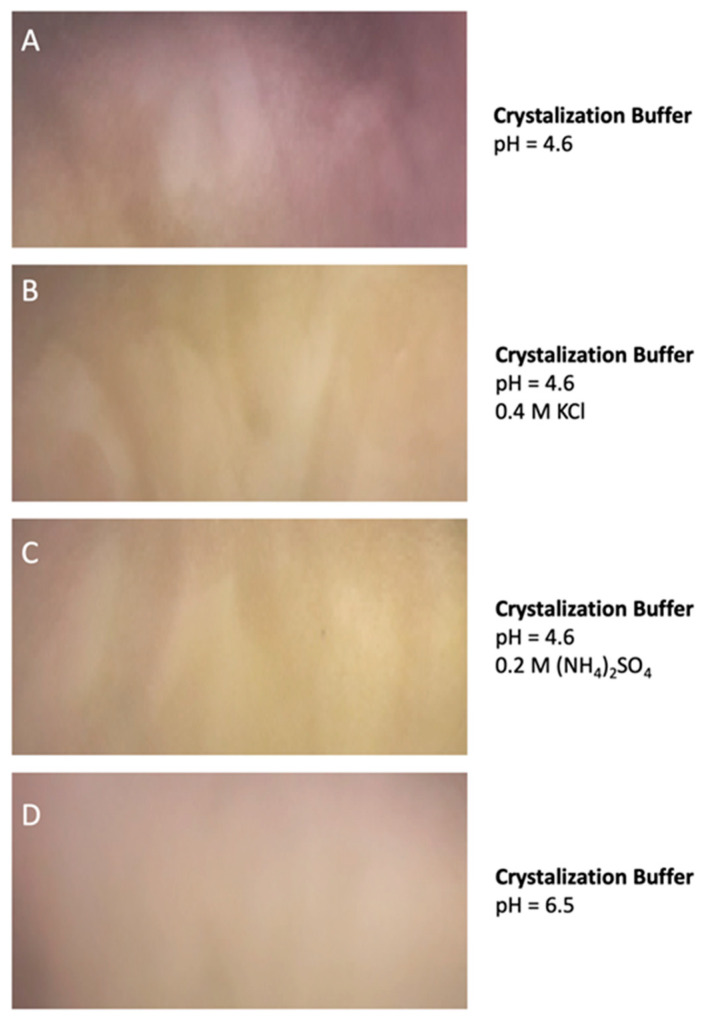
DdFXN Aggregation During Crystallization Screening Under Acidic Conditions. Four different conditions of buffer are shown: (**A**) 35% pentaerythritol propoxylate, 0.1 M acetate buffer pH 4.6, (**B**) same buffer as (**A**) but with the addition of 0.4 M KCl, (**C**) same buffer as (**A**) but with the addition of 0.2 M (NH_4_)_2_SO_4_, and (**D**) a control at pH 6.5 where there was no aggregation.

**Figure 6 ijms-21-06821-f006:**
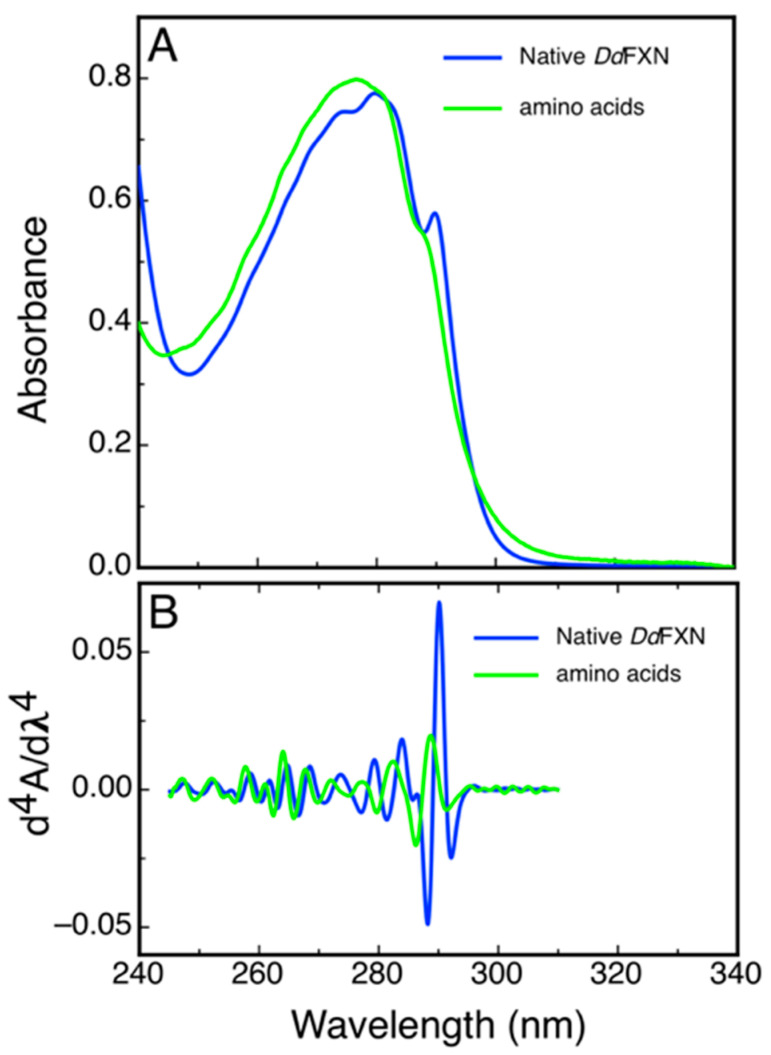
Characterization of DdFXN UV Absorption Spectra. UV absorption (**A**) and the fourth derivative spectra (**B**) corresponding to purified DdFXN (36.5- µM protein concentration, blue) or a solution containing 109.5 µM NATA, 109.5 µM NAYA, and 182.5 µM Phe (green). This composition of amino acid residues is the same (molar concentration of each residue) as in a 36.5-μM DdFXN protein solution. The contribution to the absorption of the disulfide bonds was not taken into account because Cys residues were in the reduced state, as thiols.

**Figure 7 ijms-21-06821-f007:**
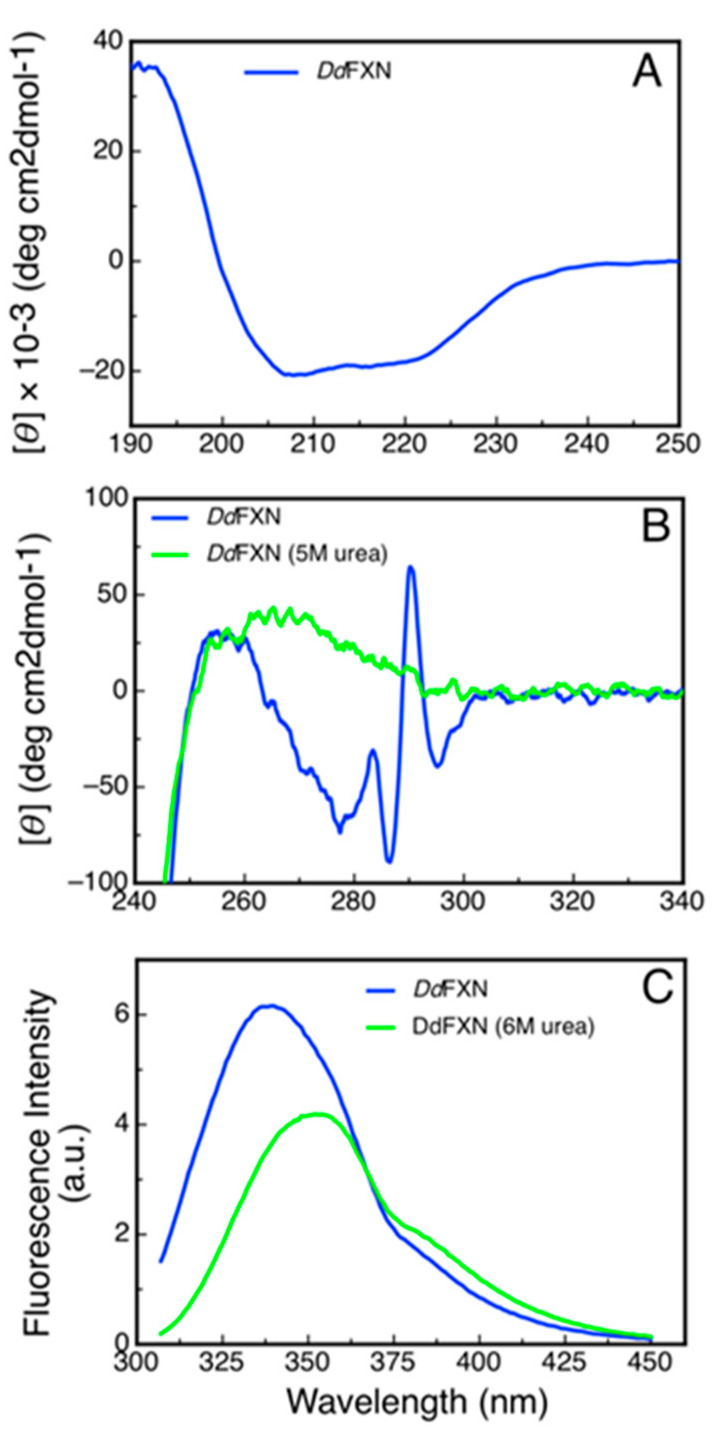
Spectroscopic Characterization of the Structure of DdFXN by Circular Dichroism and Fluorescence. (**A**) Far-UV spectrum, (**B**) near-UV spectra and (**C**) tryptophan fluorescence spectra of DdFXN. Protein concentration was 8, 30 and 3 µM, respectively.

**Figure 8 ijms-21-06821-f008:**
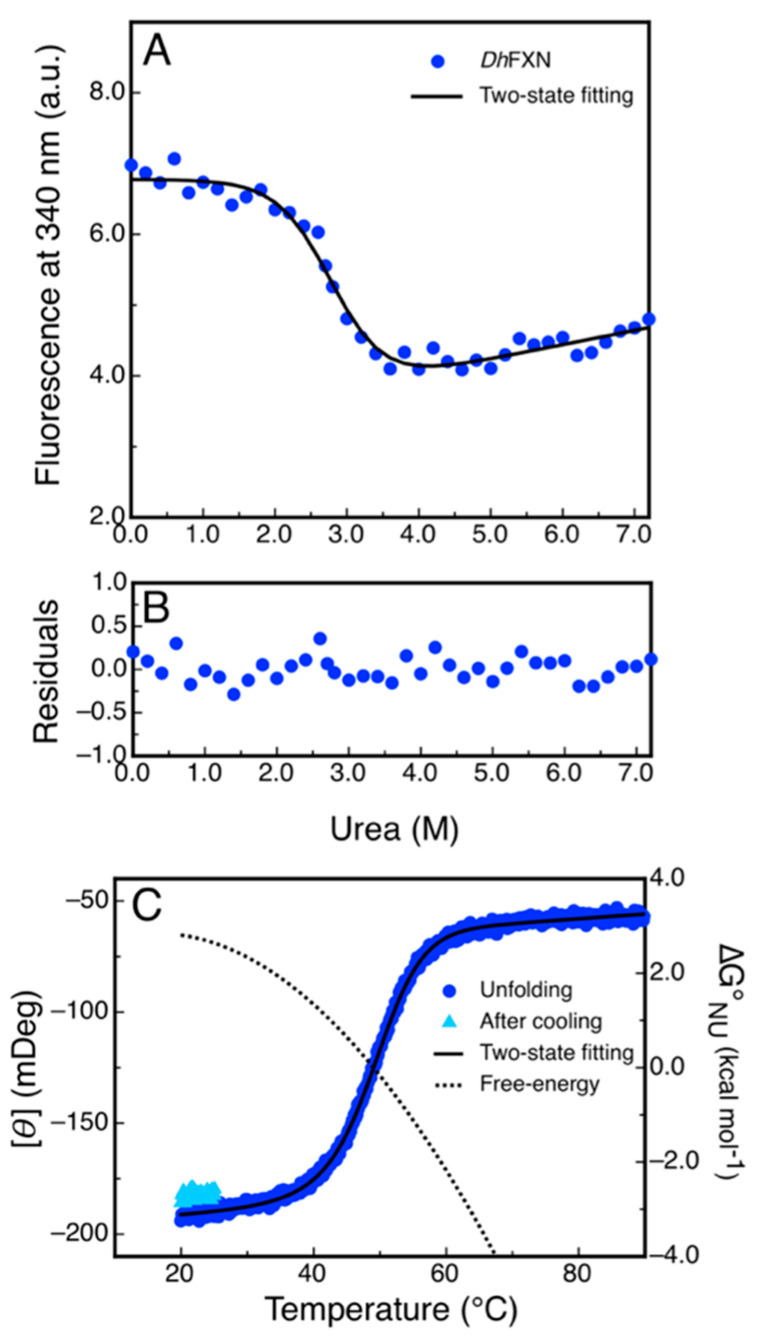
Equilibrium Unfolding of DdFXN. (**A**) Urea-induced unfolding monitored by tryptophan fluorescence at 340 nm. (**B**) Residuals of the fitting showed in A. (**C**) Temperature-induced unfolding followed by the far-UV CD signal at 220 nm. The unfolding curve is shown in blue and the signal recovered after cooling the sample and restarting the heating is shown cyan. The dotted curve corresponds to the difference in free energy calculate by means of a fitting of a two-state model to the data.

**Figure 9 ijms-21-06821-f009:**
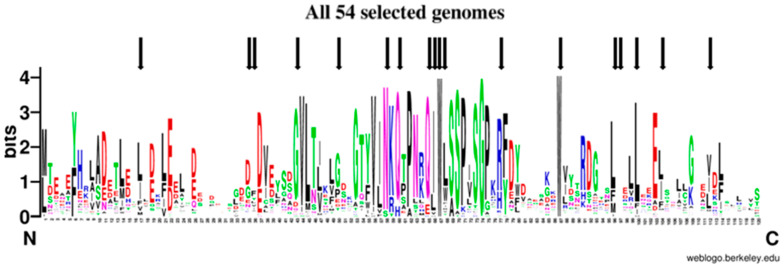
Sequence Logo Visualization of the FXN Functional Domain Alignment Across 54 Representative Genomes. FXN homologues from 54 representative genomes (Appendix A) were aligned (see materials and methods); black arrows mark positions corresponding to amino acids mutated in FRDA.

**Figure 10 ijms-21-06821-f010:**
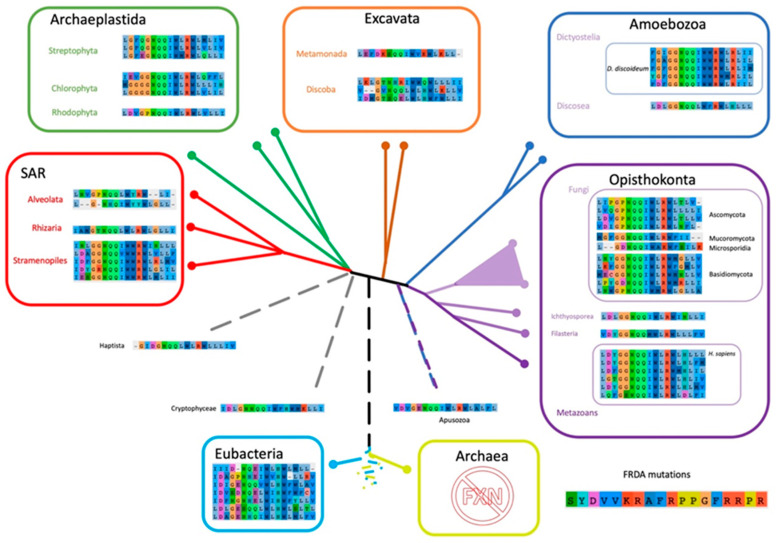
Scheme of Eukaryotic Phylogeny Showing the Corresponding Residues Involved in FRDA Across the Tree of Life. Eukaryotic phylogeny based on Adl et al. [56]. The FXN positions associated to Friedreich’s Ataxia are shown for each representative species selected in FXN phylogeny (All54G_cured_alignment.aln, Appendix A).

**Table 1 ijms-21-06821-t001:** *Dictyostelium discoideum* Iron-Sulfur Cluster Biogenesis Machinery.

Protein (Full Sequence)					Conserved Domain		
Name Description		ID Dictybase ^a^/NCBI ^b^	ID Human Orthologue ^c^	% Identity/Similarity ^d^	% Coverage/e-Value	ID (Description) ^e^	Interval/e-Value
NFS1—Cysteine desulfurase	ISU COMPLEX	DDB_G0279287/XP_641773	Q9Y697	59.9/77.6	87/0	cl18945 (Aspartate aminotransferase)	49–448/ 0
ACPM—Acyl carrier protein		DDB_G0291866/XP_629874	O14561	30.1/48	55/8.0 × 10^−15^	cl09936 (Phosphopantetheine attachment site)	2–120/ 5.49 × 10^−18^
ISCU—Fe-S cluster assembly enzyme		DDB_G0283003/XP_639309	Q9H1K1	50.2/59.4	75/1.0 × 10−^63^	PRK11325 (scaffold protein; Provisional)	56–179/ 1.51 × 10^−83^
FDX2—Ferredoxin-2		DDB_G0267486/XP_647073	Q6P4F2	36.5/51.6	62/2.0 × 10^−40^	PLN02593 (adrenodoxin-like ferredoxin protein)	44–159/ 3.75 × 10^−66^
FXN—Frataxin		DDB_G0293246/XP_629221	Q16595	30/47.2	49/4.0 × 10^−20^	pfam01491 (Frataxin_Cyay. Frataxin-like domain)	88–190/ 1.36 × 10^−43^
LYRM4 (ISD11)		DDB_G0290725/XP_635573	Q9HD34	43.9/68.1	89/1.0 × 10^−23^	cd20264 (Complex1_LYR_LYRM4, leucine-tyrosine-arginine motif found in LYR motif-containing protein 4)	7–73/ 1.42 × 10^−25^
HSPA9 (GRP75)—Hsp 70	CLUSTER RELEASE COMPLEX	DDB_G0293298/XP_629204.1	P38646/HSPA9	58.6/71.6	89/0	PRK00290/cl35085 (dnaK molecular chaperone, Provisional)	30–658/ 0
GrpE1 (MGE)—PAM complex		DDB_G0283763/XP_638912.1	Q9HAV7	38.2/54.4	61/2.00 × 10^−35^	pfam01025 (grpE)	42–209/ 1.99 × 10^−52^
GLRX5—Glutaredoxin-related protein		DDB_G0274657/XP_644145	Q86SX6	32.4/58.5	59/5.00 × 10^−29^	cd03028 (Glutaredoxin family, PKC-interacting cousin of TRX PICOT-like subfamily)	147–235/ 1.88 × 10^−56^
						cd02984 (TRX_PICOT, TRX domain, PKC-interacting cousin of TRX) subfamily)	7–105/ 2.38 × 10^−38^
HSC20 (HscB), Fe-S cluster co-chaperone		DDB_G0280889/XP_640965.1	Q8IWL3	21.3/39	80/1.00 × 10^−25^	PRK05014 (co-chaperone HscB; Provisional)	136–290/ 1.22 × 10^−24^
BolA1		DDB_G0274169/XP_644296.1	Q9Y3E2	36.8/50.7	74/1.00 × 10^−31^	pfam01722 (BolA, BolA-like protein; morphoprotein BolA from *E. coli*)	15–89/ 1.03 × 10^−35^
BolA1		DDB_G0290319/XP_635799.1	Q9Y3E2	32.9/50.7	72/2.00 × 10^−24^	pfam01722 (BolA-like protein; morphoprotein BolA from *E. coli*)	49–121/ 4.46 × 10^−40^
AbcB7 ABC transporter		DDB_G0292554/XP_629496.1	O75027	41.4/59.3	84/0	COG5265 (ATM1, ABC-type transport system involved in Fe-S cluster assembly, permease and ATPase components)	199–693/0
ISCA1	ISA COMPLEX	DDB_G0280173/XP_639260.1	Q9BUE6	32.9/55.5	82/1.00 × 10^−31^	TIGR00049 (Iron–sulfur cluster assembly accessory protein)	21–123/1.28 × 10^−39^
ISCA2		DDB_G0284809/XP_641284.1	Q86U28	23.3/37.7	77/9.00 × 10^−26^	TIGR00049 (cluster assembly accessory protein)	102–205/3.95 × 10^−35^
IBA57-Putative transferase CAF17		DDB_G0285011/XP_639996	Q5T440	24.6/37.5	64/3.00 × 10^−36^	COG0354 (YgfZ, Folate-binding Fe-S cluster repair protein possible role in tRNA modification)	13–266/3.99 × 10^−42^
NFU1—NIF system Fe-S cluster scaffold protein	LATE STAGE TRANSFER	DDB_G0285593/XP_638146	Q9UMS0	37.4/50.1	73/3.00 × 10^−64^	pfam08712 (Nfu/NifU N- terminal)	100–187/6.63 × 10^−40^
						pfam01106 (NifU carboxy-terminal)	216–282/2.11 × 10^−31^
BolA3		DDB_G0274439/XP_639996.1^+^	Q53S33	26.6/38.7	44/4.00 × 10^−13^	COG0271 (BolA Stress-induced morphogen, activity unknown, signal transduction mechanisms) *	28–99/ 1.78 × 10^−21^ *

a: ID Dictybase; b: ID source Genbank; c: ID source Uniprot; d: Similarity Matrix BLOSUM62: e: ID source conserved domains: CDD NCBI; * new gene model annotated in this work (Appendix A), + this protein is about 50 amino acids shorter on the amino terminus compared to homologs.

**Table 2 ijms-21-06821-t002:** Secondary Structure content of DdFXN.

Secondary Structure Type	Human FXN (PDB ID 1EKG) ^1^	DdFXN Model (A) ^1,3^	DdFXN Model (B) ^1,2^	DdFXN Model (C) ^1,4^	DdFXN Jpred 4 ^5^	Bestsel (200–250 nm) ^6^	Bestsel (190–250 nm) ^7^
*α*-helix	30.3	32.7	34.2	34.2	31.7	38.0	41.4
*β*-strand	30.3	35.4	34.2	35.1	24.4	22.4	15.7
Other	39.4	31.9	31.6	30.7	43.9	39.6	42.9

^1^ Calculated using Yasara software [46]. ^2^ Modeled using PDB ID: 6FCO as the template (*Chaetomium thermophilum*). ^3^ Modeled using PDB ID: 1EKG as the template (*H. sapiens*). ^4^ Modeled using PDB ID: 2EFF as the template (*E. coli*). ^5^ Calculated from secondary structure predictions using Jpred-4 [40]. ^6^ Calculated using the program Bestsel with wavelength range 200–250 nm [47,48]. ^7^ Same as ^6^ but with wavelength range 190–250 nm.

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
