# Peer review of "A Highly Conserved Iron-Sulfur Cluster Assembly Machinery between Humans and Amoeba *Dictyostelium discoideum*: The Characterization of Frataxin"

_ijms, 2020, doi:10.3390/ijms21186821_

Round 1

Reviewer 1 Report

This manuscript reports a comparative study of the Fe-S cluster assembly components of the amoeba Dictyostelium discoideum with other known systems to assess its functionality. This type of comparative studies are very powerful to pinpoint key functional amino acids, which helps elucidate complex biochemical process such as Fe-S cluster biosynthesis. The data are well-presented and focus on the very important frataxin protein. However, the most recent literature is not properly described/integrated and thus the conclusions are not sufficiently strengthened. There is an array of recent data that now provides valuable details on the role of key amino acids of NFS1, ISCU and FXN and on the functional role of the FXN protein, but this has been somehow overlooked here. I suggest that the authors provide more details about the recent literature (without being exhaustive as for a review) and properly cite the corresponding works.

Line 62-68: The authors rely on comparative study to assess the functional relevance of the Fe-S cluster assembly of the Dictyostelium discoideum amoeba, but the function of the key amino acids of ISCU are not properly described. In the light of the most recent studies, it must be mentioned that Asp37 is not only important to destabilize the Fe-S cluster but also for the assembly of the [2Fe2S] cluster per se as it is involved in initial binding of the ferrous iron that will lead to formation of the Fe-S cluster. Indeed, the following four conserved amino acids: Cys44, Asp46, Cys69 and His137 are essential to bind the ferrous iron ion (Gervason et al. Nat. Comm. 2019, Lin et al. JACS 2020), and Cys138 is the persulfide receptor (Bridwell-Rabb et al. Biochemistry 2014, Parent et al. Nat Commun 2015, Gervason et al. Nat. Comm. 2019). A correlation between binding of ferrous iron in the assembly site and Fe-S cluster assembly has been recently reported by Gervason et al. (Nat. Comm. 2019) with the mouse system, which as a eukaryotic system is highly relevant to the present study. Binding of ferrous iron in the assembly site has been later on confirmed with the E. coli system by Lin et al. (JACS 2020), which indicates that this mechanism is probably well-conserved across species. Therefore, the role of each amino acid must be more clearly specified in the manuscript. The authors may also highlight the structural flexibility of the assembly site of ISCU, as these amino acids alternatively serve as ferrous iron binders, sulfur receptor and Fe-S cluster ligands. These studies must be described in details and properly cited as they are highly relevant to the present study to pinpoint functional key amino acids of ISCU.

Line 69-71: The most recent study indicate that FXN accelerate sulfur donation to ISCU (Bridwell-Rabb et al. Biochemistry 2014, Parent et al. Nat Commun 2015, Gervason et al. Nat. Comm. 2019) and the CryoEM structure of the hetero-pentamer NFS1-ISD11-ACP-ISCU-FXN complex shows that key amino acids of FXN are pointing toward the assembly site of ISCU (Fox et al. Nat Commun 2019), which does not fit the notion of an allosteric activator. An allosteric modulator does not bind to the active site of a protein but at a distant site to regulate its activity and/or substrate specificity. The physiologically relevant reconstitution reported by Gervason et al. indicates that FXN is rather a kinetic activator of the metal-dependent persulfide transfer process to ISCU and the structure solved by Fox et al. is consistent with this model, as FXN appears to modulate the coordination sphere of the metal in ISCU. Although, the mechanism of persulfide transfer acceleration by FXN is not completely understood, these studies must be properly described and cited here; this is fully relevant for a comparative studies of conserved amino acids in Dictyostelium discoideum FXN.

Line 71-75: It is unclear why the authors focus on the putative function of FXN as an iron chaperone/donor. Moreover, reference are missing here. Do they consider that FXN is a bi-functional protein: iron chaperone and sulfur donation activator or do they consider that these functions are mutually exclusive? This must be clarified. Finally, Gervason et al. Nat. Comm. 2019 reported compelling evidence that FXN is not necessary for iron insertion in ISCU. This work should be described here too. The authors wrote that FXN is able to release iron even in the presence of DTT. However, DTT is an artificial reductant that was used in some reconstitutions instead of FDX2 to reduce the persulfide into persulfide, but this molecule cannot be considered as a surrogate of FDX2 as it operates via a very distinct mechanism by releasing free sulfide through reduction of the persulfide of NFS1. The mention of DTT must be avoided here as it leads to non-physiological Fe-S cluster assembly.

Line 108-110: References are missing. A very recent article came out which indicates that FDX2 is also involved in reductive coupling of two [2Fe2S] into a [4Fe4S] (Weiler et al. PNAS 2020). The authors were probably not aware of this work when submitting their manuscript, but they might may want to cite it, in addition to other references on this matter.

Line 184-185: Asp37 is not only involved in binding of zinc but also of iron which is more relevant to Fe-S cluster assembly than zinc.

Figure 2: The text superimposed on the structures (A, B and C) are illegible, please clarify this part. Would be nice to see the side chains two, especially Trp155 and Arg165 that form a π-cation pair in some FXN structures. Is it the case for Dictyostelium discoideum FXN? According to the CryoEM structure of the NFS1-ISD11-ACP-ISCU-FXN complex (Fox et al. Nat Commun 2019), these amino acids are critical to modulate the coordination sphere of the metal and are thus most likely crucial for the “enzymatic” activity of FXN and possibly also interaction withnthe complex.

Line 282: Evaluation of enzymatic activity of FXN variant with Kcat and KM as described in ref 33 is not physiologically relevant as it rely on reduction by DTT and thus must be avoided.

Line 338: “and” instead of “an”

Line 430: “..such the second…” a “as” is probably missing here

Line 443 and the paragraph thereafter: “Trp155 may be a key residue for persulfide transferring…”. However, as Trp155 strongly interacts with ISCU it may also be required for the interaction with the complex. The authors should strengthen their discussion with a more detailed description of the structure of the NFS1-ISD11-ACP-ISCU-FXN complex (Fox et al. Nat Commun 2019). Indeed, Trp155 pushes His137 aside from the metal centre of ISCU via a π-π interaction, which may facilitate access to the persulfide receptor Cys138 and thus could support the hypothesis that Trp155 is a key residue for persulfide transferring rather (possibly in addition to a role in protein-protein interaction).

Line 485: As pointed above “allosterically” is not suited to the role of FXN

Line 499: “…more unstable the human…” a than seems to missing

Line 540: “Whether or not DdFXN is also involved in iron maintenance and metal ion bioavailability inside the mitochondria, is a special issue that can be experimentally evaluated”. Please provide details on how this can be experimentally tested ?

Reviewer 2 Report

This paper by Olmos et al. is well written and interesting. Overall I recomend it for publication but have a few suggestions to strengthen the paper.

In the introduction the role of FXN is state to be providing metal ions for biosynthesis. Please give a reference for these statements.

Also given that metal binding/ release is the function of FXN it would be good to measure the iron affinity DdFXN and if possible show release of the metal.

Some minor comments:

1) It would be helpful to bring Figure 1 earlier in the text (into the introduction section) as this will help the reader visualise the role of FXN.

2) To the untrained eye it I cannot see any aggregation or difference between the images in Figure 5. Please can you annotate them or provide improved figures.

Reviewer 3 Report

The authors here report on the proposed use of the Dictyostelium Discoideum amoeba as a biological model to study iron sulfur cluster biosynthesis at the molecular, cellular and organism level. Indeed, the authors highlighting of the potential variation patterns between humans and ameoba reflecting functional importance for protein positions is of particular interest as it supports the initial hypothesis of shared and conserved iron assembly machinery between both species. 

Overall, the authors have provided a comprehensive evaluation and assessment of the study and indeed, and this work does open new avenues for studying the physiopathology of this metabolic pathway. Especially regarding the mutational manipulation of this super-complex to enhance our understanding for several sever associated human diseases. 

Round 2

Reviewer 2 Report

The manuscript is now suitable for publication.